biomaterials/materials science

alginate fibres, microfluidic spinning, characterization, grooved structure

**Author for correspondence:**
Bingyao Deng
e-mail: bydeng168@163.com

This article has been edited by the Royal Society of Chemistry, including the commissioning, peer review process and editorial aspects up to the point of acceptance.

# Facile fabrication and characterization on alginate microfibres with grooved structure via microfluidic spinning

Xiaolin Zhang[1], Lin Weng[2], Qingsheng Liu[1], Dawei Li[1] and Bingyao Deng[1]

[1]Laboratory for Advanced Nonwoven Technology, Key Laboratory of Eco-Textiles, Ministry of Education, Jiangnan University, Wuxi 214122, People's Republic of China
[2]Okinawa Institute of Science and Technology, Okinawa 904-0495, Japan

XZ, 0000-0003-0238-6798

Alginate microfibres were fabricated by a simple microfluidic spinning device consisting of a coaxial flow. The inner profile and spinnability of polymer were analysed by rheology study, including the analysis of viscosity, storage modulus and loss modulus. The effect of spinning parameters on the morphological structure of fibres was studied by SEM, while the crystal structure and chemical group were characterized by FTIR and XRD, respectively. Furthermore, the width and depth of grooves on the fibres was investigated by AFM image analysis and the formation mechanism of grooves was finally analysed. It was illustrated that the fibre diameter increased with an increase in the core flow rate, whereas on the contrary of sheath flow rate. Fibre diameter exhibited an increasing tendency as the concentration of alginate solution increased, and the minimum spinning concentration of alginate solution was 1% with the finest diameter being around 25 μm. Importantly, the grooved structure was obtained by adjusting the concentration of solutions and flow rates, the depth of groove increased from $278.37 \pm 2.23$ μm to $727.52 \pm 3.52$ μm as the concentration varied from 1 to 2%. Alginate fibres, with topological structure, are candidates for wound dressing or the engineering tissue scaffolds.

# 1. Introduction

To the best of our knowledge, it is vital that cells grow on tissue following controlled behaviour. Hence, the materials

systematically guiding and organizing the cellular behaviour is essential for successful regenerative medicine and tissue engineering scaffolds [1–3]. Cellular behaviour including proliferation, adhesion, migration and differentiation is affected by the substitute which could be fabricated by a variety of materials, mimicking the extracellular matrices (ECMs) [4–6]. Furthermore, the substitute plays a crucial role in the construction of multifunctional tissue and organs [7,8]. As reported, various strategies have been used to fabricate man-made ECMS, aiming at controlling the cellular behaviour. For this purpose, ECMs could be designed by the functional raw materials or the specific surface morphology to facilitate cell formation and growth [9,10]. Three-dimensional topography of the micro/nanometre scale could be constructed on the scaffold, which would contribute to engineering and organizing the cellular behaviour [11–14]. However, it is a great challenge to fabricate the ECMs on the tissue scaffolds or biomaterials, which need to address lots of problems.

Currently, diverse methods have been developed in efforts to fabricate the topography of micro/ nanostructure for promoting the cells well aligned and organized [15,16]. Micro contact printing technology has emerged as a powerful platform to imitate ECMs, however, limited to two-dimensional environments [17]. Electrospinning has been widely reported to fabricate three-dimensional materials to pattern ECMs, but of limiting applications because of the poor mechanical performance and low efficiency [18,19]. Freeze-drying has been adequately employed to fabricate the materials with porous micro/nanometre structure, whereas exposing the restricted application on account of the hard and brittle profile [20,21]. Topological morphology could be achieved by the micro-electromechanical systems, including the fabrication of diverse parallel grooves with different depth [22]. However, the complicated process, high costs and limitations on the available materials impede the extensive applications. As a solution, microfluidic spinning technology is a growing explosion in the fabrication of materials with diverse micro/nanometre-scale topological structure and specific function. Microfluidic spinning technology exhibits the wide adaptability for all sorts of polymers and offers a straightforward platform for the facile fabrication of biomaterials with tunable morphological, structural and chemical features [23,24]. Therefore, tremendous interest in microfluidic spinning [25–29] is evoked in pioneering new building biomaterials towards the development of functional materials with robust properties.

Herein, a simple strategy was presented based on the microfluidic spinning technology (MST) for the facile fabrication of alginate fibres. The microfibres with grooved structure from the polymer in one step by MST were constructed and the formation mechanism of grooves was investigated. In addition, the micro/nanometre grooved structure with various scale on the fibres was achieved by adjusting the spinning parameters. In the view of the scarce reporting on this study, the inherent properties of polymer including rheological behaviour and crystalline status were investigated, which was devoting to revealing the condition of the formation of fibre. Furthermore, the characterization on the fibrous surface morphology and topography was employed to illustrate the work. The alginate microfibres, with topological structure, have the candidate for wound dressings and tissue scaffolds, by virtue of its biocompatibility and non-toxicity.

# 2. Material and methods

## 2.1. Fabrication of grooved microfibres

The alginate microfibres were fabricated by the microfluidic spinning device (Janus New-Materials Co., Ltd, Jiangsu, China) composing of a coaxial flow of two solutions. Various concentration of alginate solution (1%, 1.5%, 2% (w/v)) were prepared by dissolving sodium alginate powder (Sigma-Aldrich, St. Louis, MO, USA) in deionized water as the core flow. The sheath flow consisting of 3% (w/v) $CaCl_2$ (Sinopharm Chemical Reagent Co., Ltd, China) in ethanol (Titan Scientific Co., Ltd, Shanghai, China) was used to induce the instant gelation of alginate solution for the generation of fibre. A microfibre was extracted from the dressings, which was conducive to investigate the influence of spinning parameters on the fibres with grooved structure.

## 2.2. Rheological measurement

Rheological properties of alginate solutions were investigated by Physica MCR301 Rheometer (Anton Paar, Austria) fitted with plate PP-50 geometry (50 mm diameter), having a temperature control device to maintain the measuring temperature at 25°C. Aqueous system of alginate solution with

various concentrations was placed between the parallel-plate geometry with a gap of 1 mm. Steady-state shear and dynamic oscillatory shear modes were employed to study the flow properties and viscosity behaviour of the samples under steady-state and dynamic loading conditions, respectively. Resultant viscosity ($\eta$) and shear stress ($\tau$)-dependence curves of shear rate ($\gamma$) were studied from 0.01 to 100 s$^{-1}$ to characterize the viscoelasticity of the polymer. Dynamic strain sweep tests were carried out from 0.01 to 1000% with a constant frequency of 10 Hz. Sweep frequency dependence of elasticity modulus (G′) and viscosity modulus (G″) were monitored from 0.1 to 100 Hz.

## 2.3. Scanning electron microscopy

The difference in morphologies of fibres fabricated with distinct condition was observed using scanning electron microscopy (SEM). In order to keep the original morphology for all the samples before SEM imaging, the samples were washed and dehydrated by the ethanol solutions, then dried at room temperature and fixed on conductive carbon tape and sputter-coated with a thin gold layer. The prepared samples were examined by SEM (SU1510, Hitachi, Japan) at an accelerating voltage of 5 kV.

## 2.4. Fourier transform infrared spectroscopy

To investigate the difference in structure of alginate polymer and fibres, FT-IR spectrum was performed via a universal ATR diamond accessory of Nicolet Is 10 (Thermo Fisher, USA). Transmission and ATR spectra were recorded with spectral region between 800 and 4000 cm$^{-1}$.

## 2.5. X-ray diffraction studies

To make a comparison of crystal structure of alginate polymer and fibres, XRD analysis was conducted on a D2 PHASER X-ray diffractometer (Bruker-AXS, Germany) equipped with Cu K$\alpha$ radiation ($\lambda = 1.5418$ Å). The one-dimensional X-ray diffraction patterns with the intensity curves, and a function of $2\theta$, were obtained from integrating the two-dimensional scattering patterns of the samples. The angle of diffraction varied from 5° to 60° was aimed at identifying the difference in the crystal profile of alginate polymer and fibre.

## 2.6. Atomic force microscopy imaging

A MultiMode8 atomic force microscope (AFM, Bruker, Germany) equipped with a Nanoscope V (NSV) controller and E scanner was used to image the morphology and topography study for samples in ScanAsyst mode. This was done by a silicon nitride cantilever with the force constant of 0.5 N m$^{-1}$ at scan rate of 5 Hz and scan line of 512. The SNL-A tip with a resonance frequency of 80 kHz and a nominal length of 125 μm was employed to work at a drive amplitude of 100.00 mV. Imaging was carried out in the air at room temperature and humidity using freshly cleaved mica as a substrate. All the measured samples were preparing by being dried and stuck to the surface of mica.

## 2.7. Statistics

All the data were analysed by Student's $t$-test to study the statistical difference of the samples, considered significant at $p < 0.05$.

# 3. Results and discussion

## 3.1. Facile fabrication of microfibres by microfluidic spinning

Alginate microfibre could be obtained because of the ion-exchange behaviour between sodium ion of alginate macromolecular chain and divalent cation, resulting in a rapid gelation to extract the fibre. The preparation of alginate fibre in Y-shaped microfluidic spinning device was typical, as exhibited in figure 1. Alginate solution as the core flow was pumped into the microchannel of microfluidic platform via a syringe, while the sheath flow composed of CaCl$_2$ solution was injected into the connection region between the inner microchannel and outer stainless needle hose. The cross-linking was triggered instantly when the Ca$^{2+}$ diffused into the alginate solution, causing the solidification

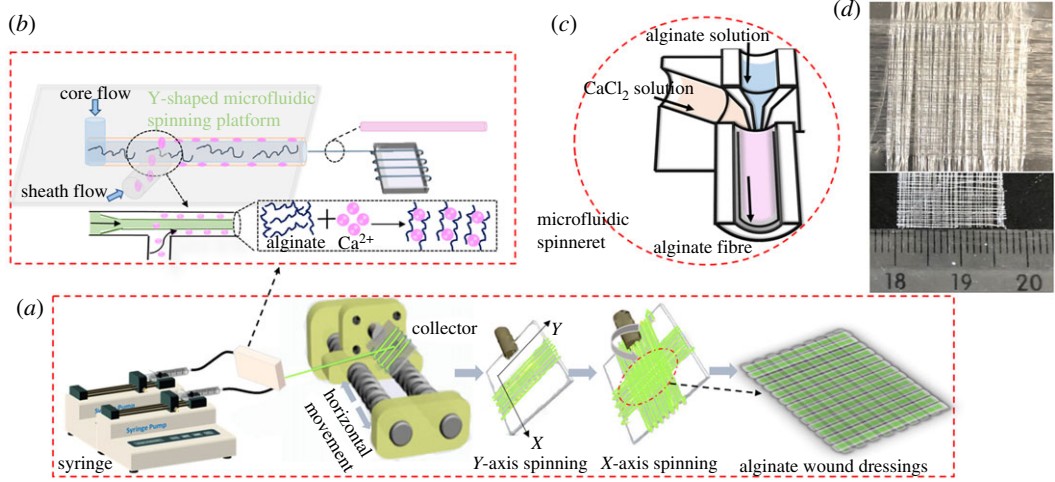

**Figure 1.** Fabrication of fibre via microfluidic spinning (*a*), the schematic of Y-shaped MST device (*b*), the drawing of microfluidic spinneret (*c*), the stereogram of alginate biomaterials (*d*).

and extraction of microfibre at the end of the outlet. As shown in figure 1*c*, alginate solution and $CaCl_2$ solution were orderly pumped into syringe, PTFE tubes and microfluidic channels, then both solutions in the end of micro-channels would meet to form the fibres. That was the result of chelation between the carboxyl group of alginate molecular chain and the calcium ion of $CaCl_2$ solution to make alginate form the 'egg-box' structure. The buckled chain of guluronic acid units acting as a two-dimensional analogue of corrugated egg-box with interstices in the calcium ions packed and coordinated to form fibre. Alginate fibres were fabricated via the microfluidic spinning technology, as displayed in figure 1. Spinning solution consisted of alginate solution as the core flow and $CaCl_2$ solution acted as the sheath flow, which were, respectively, loaded into two syringes (10 ml) mounted with 23G and 17G stainless steel needles. The diameters of core and sheath channels were 340 μm and 490 μm, respectively. A clean glass as a collector was installed in the stepmotor with the controlled r.p.m., as closely as possible to the needle. Finally, both flow rate of spinning solution and the motor were fixed at 2.5 ml min$^{-1}$ and 500 r.p.m. under a constant temperature and voltage, respectively. The fibre was fabricated by the ion-exchange behaviour of Ca−Na, resulting from the cross-linking of alginate chain and then the aggregation of guluronic acid unit. Firstly, fibre was collected orderly in the direction of X-axis by adjusting the speed of horizontal movement of collector and then for Y-axis, inducing the cross-linking and intermolecular interaction of fibres to form the alginate dressings.

## 3.2. Rheological behaviour of the alginate polymer

Obviously, the concentration of alginate polymer affected the viscosity of solution and the aggregation of macromolecular chain, which was speculated to make a large influence on the fabrication. Thus, the rheological behaviour of alginate polymer with diverse concentration was investigated as follows.

Steady-state shear and dynamic oscillatory modes were employed to investigate the rheological properties of alginate polymer of various concentration. Under rotary shear mode, the shear stress and viscosity with shear rate ranged from 0.01 to 100 s$^{-1}$ were measured as exhibited in figure 2. The shear stress and viscosity were proportional to the concentration of polymer, and the shear stress of each curve displayed a linear dependence relation with the shear rate, except for 0.5% alginate solution. For the viscosity dependence of shear rate as illustrated by the curves, the viscosity of polymer decreased with the increase in shear rate, which indicated that the alginate polymer solution had a shear-thinning behaviour. Furthermore, the polymer that had viscosity less than 1% had an abruptly decrease at a small shear rate range and then trended towards a steady state, while that of samples over 1% mainly formed a plateau.

Storage and loss modulus as a function of the sweep frequency and strain are demonstrated in figure 3*a*,*b*, respectively. It could be concluded that the loss modulus (G″) was more stable than storage modulus (G′) for all the measured region of sweep frequency and strain, meaning the viscidity of alginate polymer was greater than that of elasticity. As the dependence of strain amplitude sweep with storage modulus exhibited in figure 3*b*, the sample with higher concentration revealed the bigger

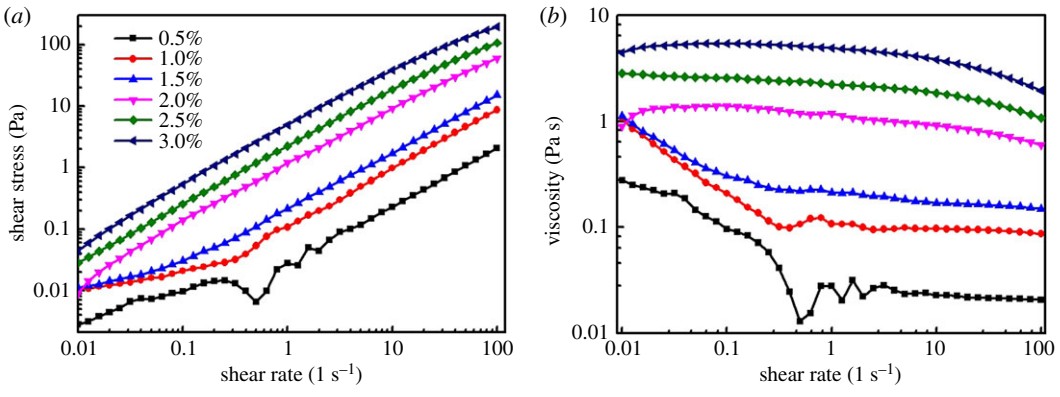

**Figure 2.** (*a*,*b*) Flow curves of sodium alginate solutions with various concentration in the steady-state mode.

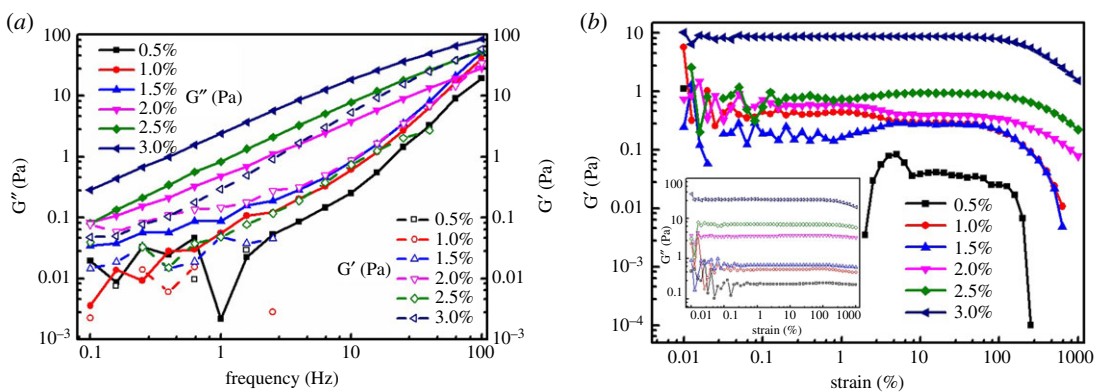

**Figure 3.** Evolution of storage modulus (G″) and loss modulus (G′) as a function of the sweep frequency (*a*) and strain-dependence curves of dynamic storage and loss modulus for sodium alginate solutions (*b*).

storage modulus. The storage modulus for each sample exhibited a plateau within the 100% strain, while showing a sharp decreasing tendency exceeding the threshold level. The loss modulus for each sample held a stable trend with increasing strain, being inferred from the inset curves of figure 3*b*. From figure 3*a*, the loss modulus of each sample showed a slight increase as the angular frequency increased. Additionally, the loss modulus of each sample dominated the whole testing area of angular sweep frequency, which implied that the sample performed a liquid state all the time. To our knowledge, the rheological behaviour of polymer represented the internal behaviour of molecules. For all the data curves, it was revealed that the concentration more than 1% of alginate polymer demonstrated a similar rheology patterns, whereas on the contrary for 0.5%.

## 3.3. The concentration and flow rate impacted on morphology of microfibre

The rheological behaviour of polymer with diverse concentration offered a new insight into the fabrication and morphological structure of fibres. Therefore, many groups of spinning concentration were selected to prepare alginate fibres. Meanwhile, morphological structure of fibres fabricated by various alginate concentrations were investigated as displayed in figure 4, with core and sheath flow rates being 1.5 and 3 ml min⁻¹. It was noticeable that an increase in the alginate solution concentration significantly increased the diameter of the fibres, while the alginate concentration of 0.5% being illustrated to generate string-of-beads fibres with different dimension, which hindered the formation of fibres. Furthermore, the diameter of fibre increased from $21.31 \pm 0.95\,\mu$m to $41.12 \pm 1.51\,\mu$m as the concentration varied from 1 to 3%, as shown in figure 4. Moreover, morphological structure of fibres fabricated by diverse flow rates were investigated as displayed in figure 5, with the concentration of alginate and $CaCl_2$ solution being 2 and 3%. It was obvious that the diameters of fibres exhibited a large variation by varying flow rates, whereas the velocity less than 0.75 ml min⁻¹ was unsuitable for fabricating fibre. The diameter of fibre increased from $31.98 \pm 0.83\,\mu$m to $47.72 \pm 1.15\,\mu$m as the core and sheath flow rates varied from 1.5 to 6.5 ml min⁻¹, as shown in figure 5.

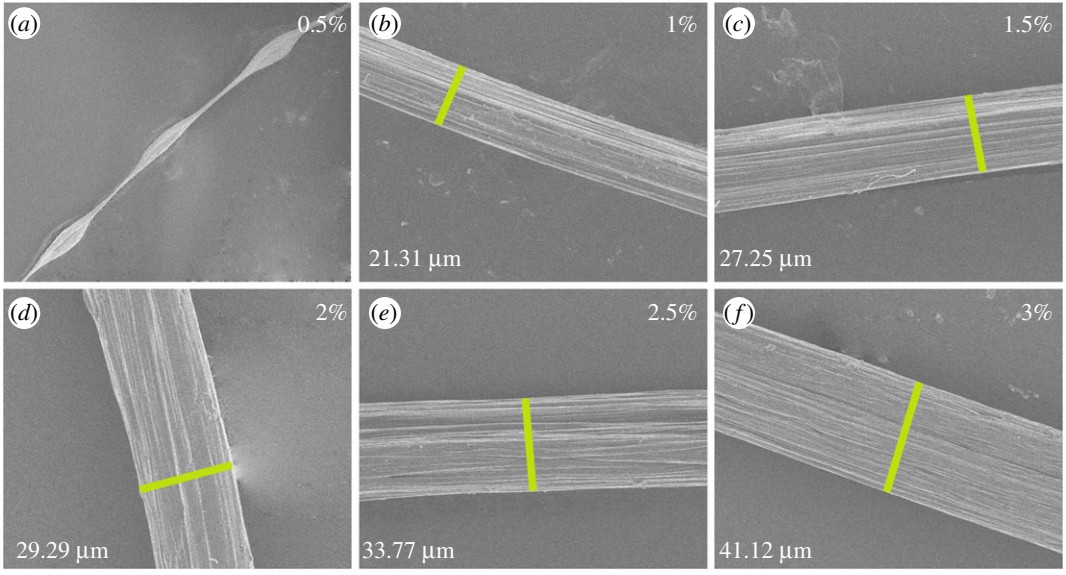

**Figure 4.** ($a$–$f$) Surface morphology of fibres fabricated by various alginate concentration.

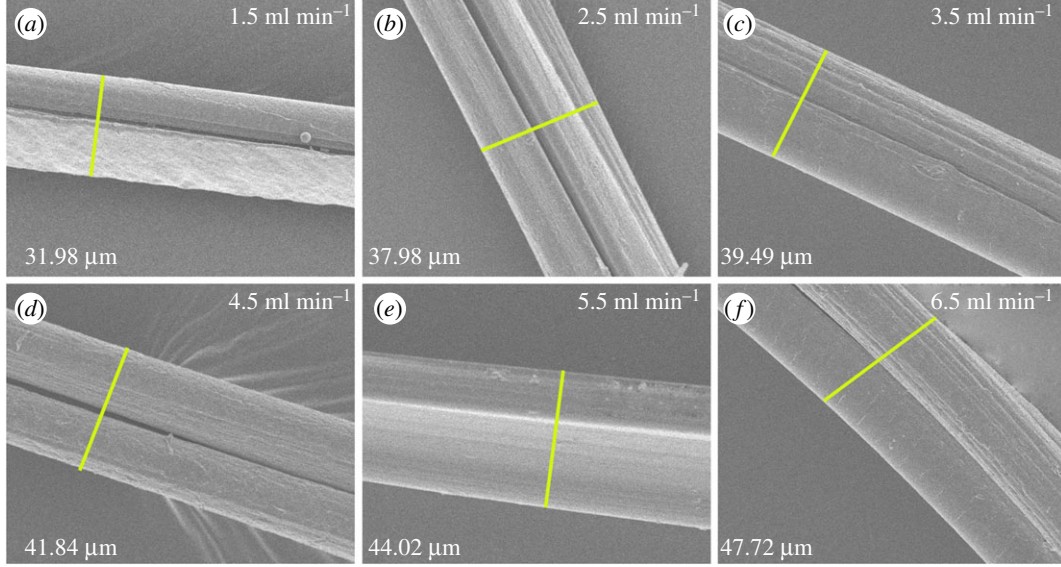

**Figure 5.** ($a$–$f$) Surface morphology of fibres fabricated at various flow rates.

To further explore the relationship of flow rate with fibre dimension, the dependence of fibre diameter on core and sheath flow rate were investigated, while the sheath and core flow rates were fixed at 3 and 1 ml min$^{-1}$, respectively, as exhibited in figure 6. It was observed that fibre diameter increased with the increase in the core flow rate, whereas on the contrary for sheath flow rate. Hence, the morphology structure of alginate fibre clearly demonstrated that polymer concentration and flow rates not only affected the fibrous diameter, but also acted as the key parameter to the processing.

In addition, the cross-section morphology of fibres fabricated by various concentrations of alginate solution and flow rates were investigated. As shown in figures 7 and 8, the cross-section of fibres was circular with a sunken segment and the grooves could be observed, which was affected by the alginate concentration and flow rates. The dimension of grooves increased with an increase in the alginate solution and flow rates under a certain condition, and then exhibited a decreased tendency. The bigger alginate concentration and flow rates improved the rate of double diffusion, resulting in the clear grooves. However, fibres formed quickly with the concentration and flow rates being high, impeding the double diffusion. That was not beneficial to the formation of grooves, corresponding to the grooves change displayed in the cross-section images of figures 7 and 8.

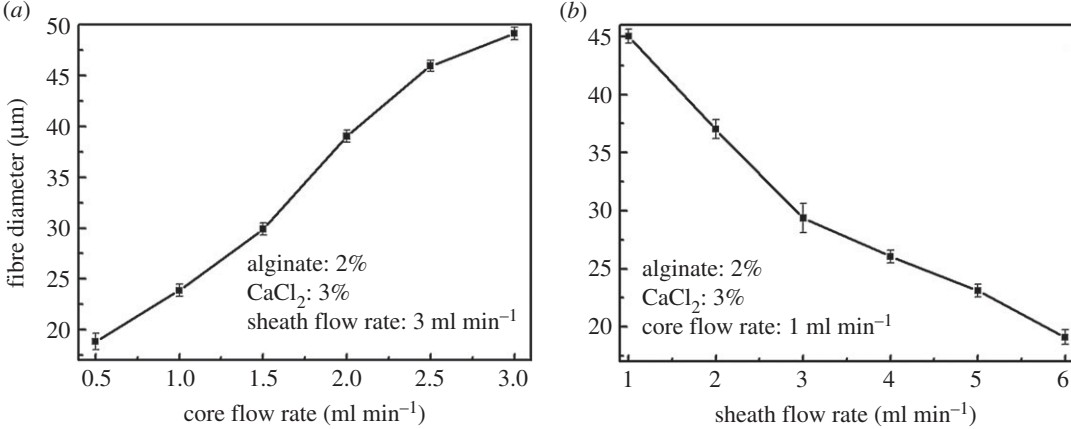

**Figure 6.** (*a*) Dependence of fibre diameter on core flow rate and (*b*) sheath flow rate.

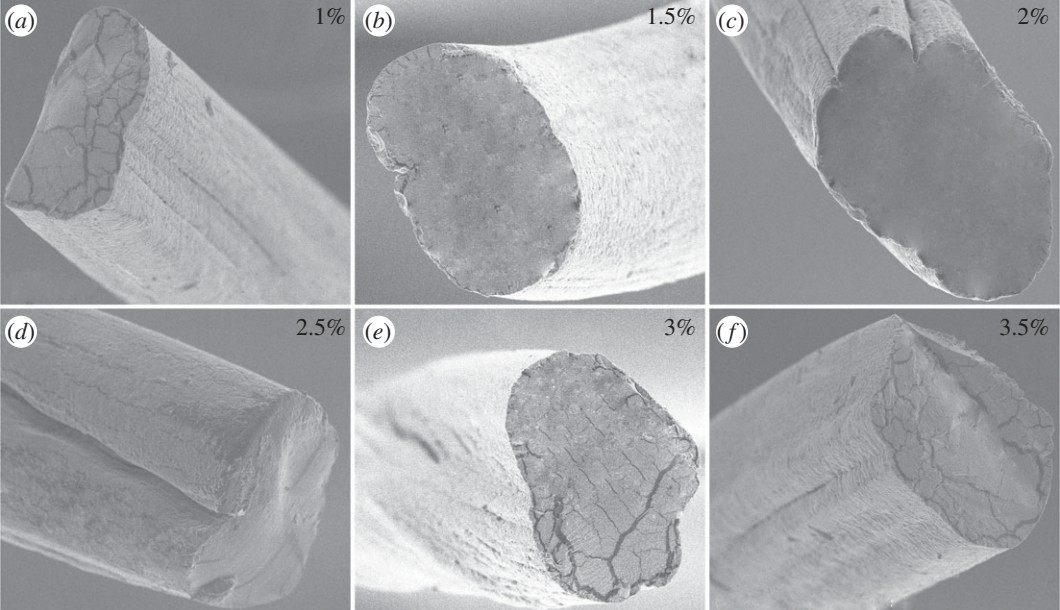

**Figure 7.** (*a*–*f*) Cross-sectional images of fibres fabricated by various alginate solutions.

## 3.4. Fourier transform infrared spectroscopy

As the research depicted, the concentration of alginate solution impacted on fibre diameter. In order to perform the further study, the concentrations of 1, 1.5 and 2% alginate solution were selected to fabricate fibres, the flow rates of core and sheath were 1.5 and 3 ml min$^{-1}$, respectively. The difference in chemical group of alginate polymer and fibre fabricated by diverse spinning concentration were conducted by the FTIR. As shown in figure 9*a*, it could be found that the alginate polymer and fibre exhibited similar absorption bands without additional peaks or noticeable shifts. A slight variation occurred at the position of a few wavenumbers and the strength of the absorption band. The peak of alginate polymer appearing at 3438 cm$^{-1}$ was assigned to vibration of the hydroxyl groups (OH) while that of fibre being 3347 cm$^{-1}$. The formation of fibre prompted the cross-link between –COOH and Ca$^{2+}$ to trigger the association of oxygen atom and calcium ion, which weakened the hydrogen bonding. Due to the generation of egg-box structure, the vibration of hydroxyl groups moved to a lower wavenumber and the region of absorption band extended. In addition, the egg-box structure in fibre impeded the C–H stretching vibration at 2919 cm$^{-1}$, resulting in the weaker absorption peak than polymer. The band of fibre displayed at 1606 and 1417 cm$^{-1}$ belonged to asymmetric and symmetric –COO stretching vibrations respectively, whereas that of alginate polymer appeared at 1606 and 1420 cm$^{-1}$. Consequently, there was no formation of new functional chemical groups after the alginate

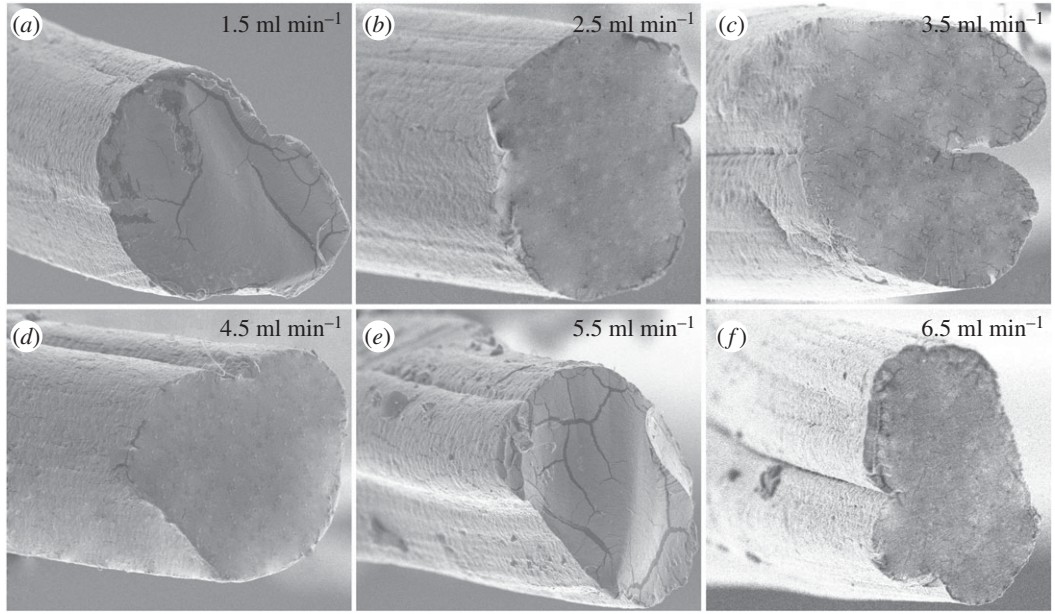

**Figure 8.** (*a–f*) Cross-sectional images of fibres fabricated at diverse flow rates.

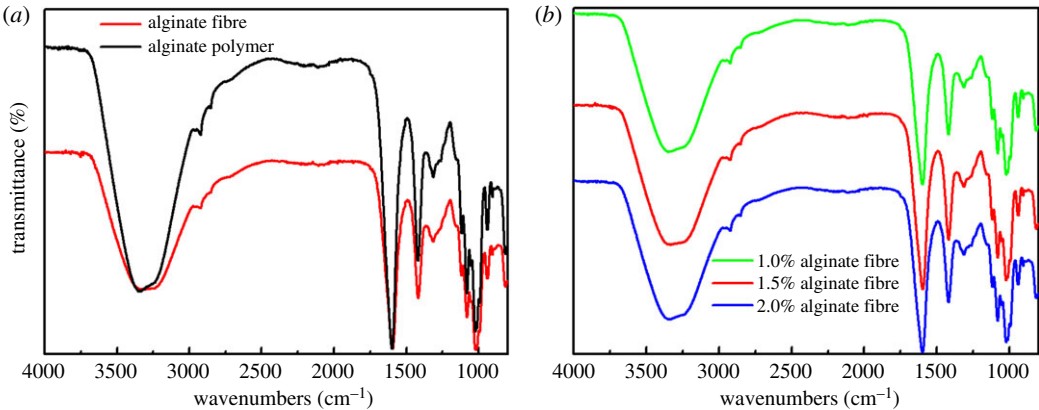

**Figure 9.** (*a*) FTIR spectra of alginate polymer and fibre and (*b*) fibres fabricated by various concentrations of alginate solution.

polymer fabricated was into the fibre. FTIR spectra of fibres prepared by alginate solution with the concentrations of 1, 1.5 and 2% were acquired in figure 9*b*. There were no noticeable variations in the intensity of absorption band and the position of wavenumbers for the fibres with diverse concentration, indicating the spinning concentration displayed little influence on the chemical group.

## 3.5. X-ray diffraction

The research on one-dimensional X-ray diffraction patterns was to evaluate the difference in the crystalline profile of alginate polymer and fibre. Moreover, the crystal structure of fibres fabricated by alginate solution with concentrations of 1, 1.5 and 2% was also investigated. It could be obtained that the alginate fibre and polymer exhibited a similar XRD pattern, as shown in figure 10*a*. The peak in both samples at $2\theta = 23.60$ in the spectra and the strength of $2\theta$ pattern indicated the degree of crystallinity. The alginate fibre provided stronger and sharper diffraction peaks, indicating the higher crystallinity and a more ordering structure compared to the polymer. The tangled roll structure of polymer turned to an ordered straight chain structure during the fabrication of fibres, resulting in the shrink of amorphous region. However, the crystal structure of three types of alginate fibre prepared by the spinning solutions of 1, 1.5 and 2% demonstrated a similar behaviour from figure 10*b*, illustrating that the concentration had little effect on the crystal profile.

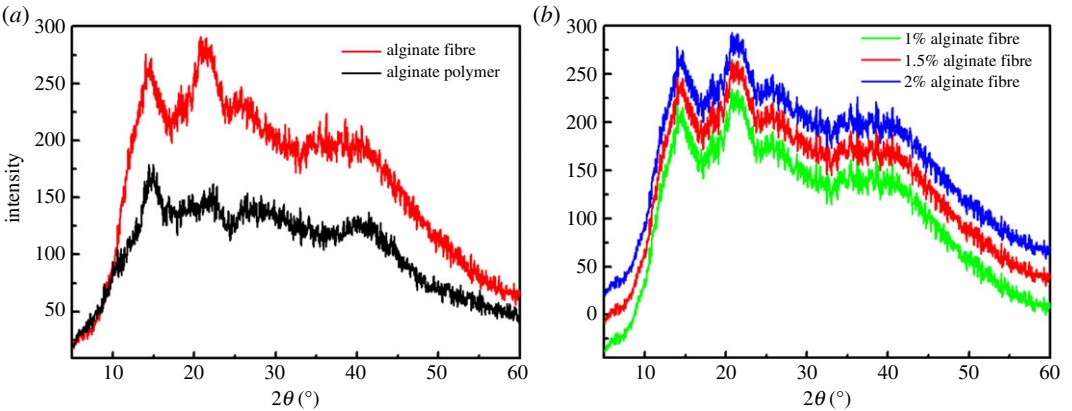

**Figure 10.** (*a*) The XRD pattern of alginate polymer and fibre and (*b*) fibres prepared by diverse concentrations of alginate solution.

## 3.6. Characterization of the diameter and surface structure of the fibres

The surface structure and diameter distribution of fibres fabricated by 1, 1.5 and 2% alginate solutions were analysed by SEM and Image-Pro Plus software, as described in figures 11 and 12. It could be clearly observed that the surface of all the fibres exhibited grooved structure and the fibre fabricated with higher alginate concentration demonstrated bigger depth and width of grooves, as depicted in figure 11. The rate of ion exchange between $Ca^{2+}$ and $Na^+$ was proportional to concentration of polymer, impacting on the formation of fibres and grooves. The higher concentration resulted in the larger ion-exchange rate, inducing the faster forming process of grooves and then generating the bigger dimension of grooves. The depth and width of groove on the fibre exhibited an expectant increase trend as the spinning concentration varied from 1 to 2%. To further quantize the dimension of groove, three-dimensional topological structure on the fibres were investigated via the AFM.

Furthermore, the diameter distribution of fibres fabricated by the concentration of 1, 1.5 and 2% alginate are exhibited in figure 12. It could be found that the diameter of 1% alginate fibre was chiefly between 23 and 26 µm, whereas that of 1.5% alginate fibre was mainly around 27 µm. For 2% alginate fibre, the diameter was widespread, ranging from 29 to 32 µm. It could be concluded that the diameter distribution of fibres was proportional to the alginate concentration. Overall, the fibre diameter exhibited an increasing tendency with increasing the alginate polymer solution.

## 3.7. AFM imaging analysis

Alginate fibres were dried and fixed on the sheet iron for subjecting to the AFM analysis to characterize the topological structure. The width and depth of groove on fibres fabricated with the concentration of 1, 1.5 and 2% were measured by the ScanAsyst mode of AFM, respectively, as shown in figure 13. And the cross-sectional structure of fibres with grooves were scanning by SEM, as displayed in figure 14. What could be caught in these images was that, although the groove widths were fairly presented at the top, at deeper levels the grooves became concave or even V-shaped. Although a little difference in some sunken or even nanogrooves appeared at the surface, the pattern of grooves was similar. The work aimed at measuring the groove depths and widths to investigate how the alginate polymer concentration impacted on the microgrooves. The microgroove depths of 1 and 1.5% alginate fibre were $278.37 \pm 2.23$ and $683.69 \pm 3.18$ nm, respectively, while that of 2% alginate fibre was $727.52 \pm 3.52$ nm. Furthermore, the widths of groove for 1, 1.5 and 2% samples were $251.33 \pm 1.67$, $419.67 \pm 2.82$ and $730.67 \pm 3.12$ nm, respectively. The results of AFM illustrated the concentration of polymer impacted on the surface morphology and the dimension of groove, to which was devoted the further study of how three-dimensional topography impacted on the cellular behaviour.

## 3.8. The formation mechanism of grooved structure

It was demonstrated that there were grooves on the fibres during the formation of spun process. Furthermore, the dimension of grooves exhibited an increasing tendency when the concentration of spinning solution increased. However, the research on the formation mechanism of grooves was

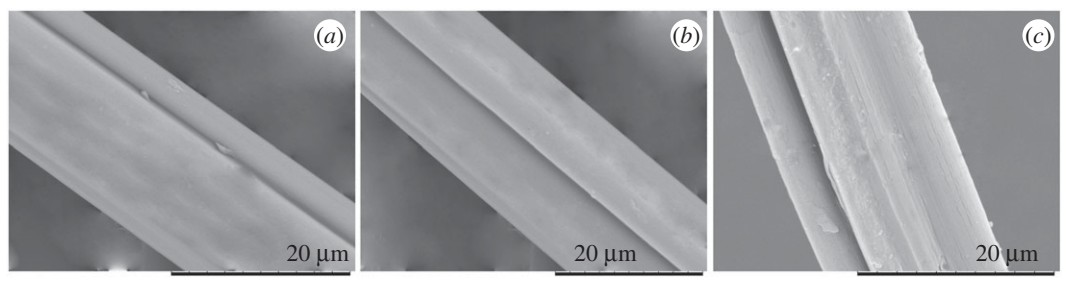

**Figure 11.** SEM image of groove with diverse width and depth on the fibre spun by (*a*) 1%, (*b*) 1.5% and (*c*) 2% alginate solutions.

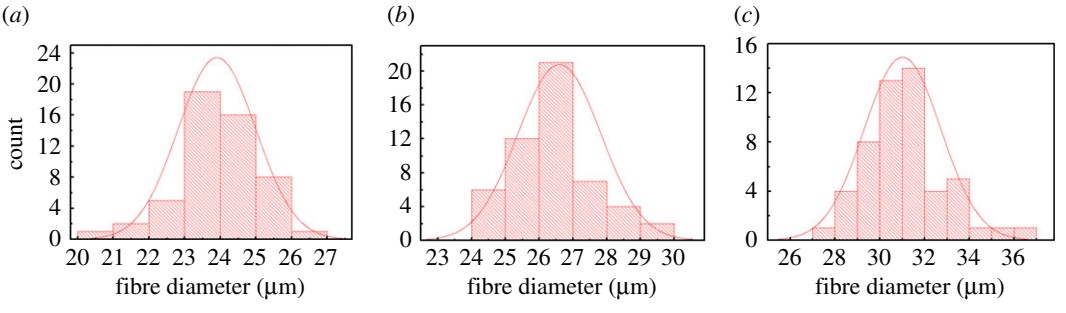

**Figure 12.** Diameter distribution of fibres spun by (*a*) 1%, (*b*) 1.5% and (*c*) 2% alginate solutions.

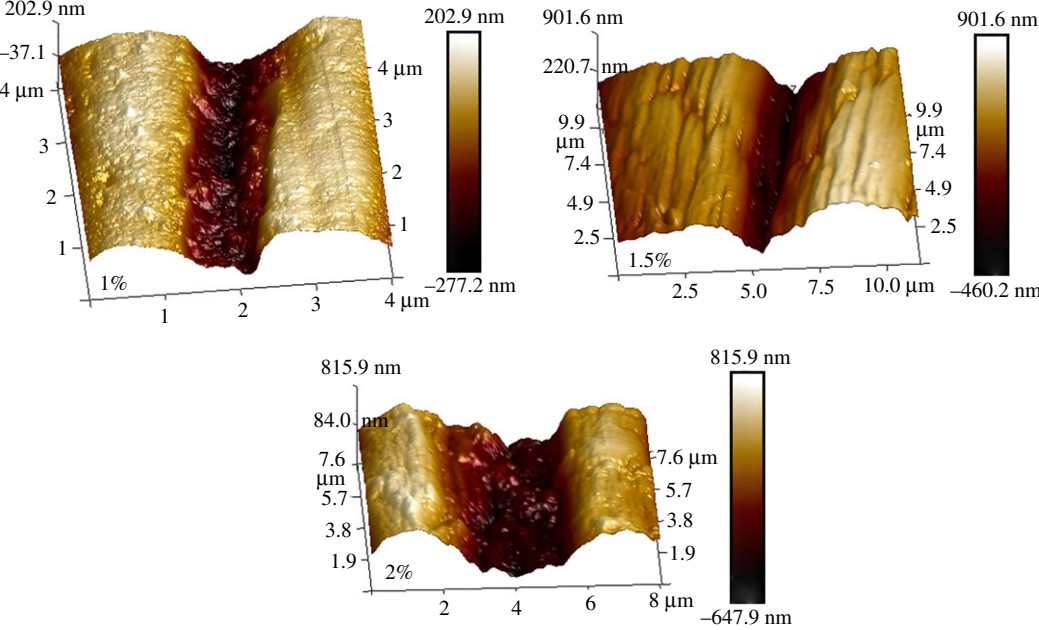

**Figure 13.** AFM graphs of topographies on the alginate fibre spun at the concentrations of 1, 1.5 and 2%.

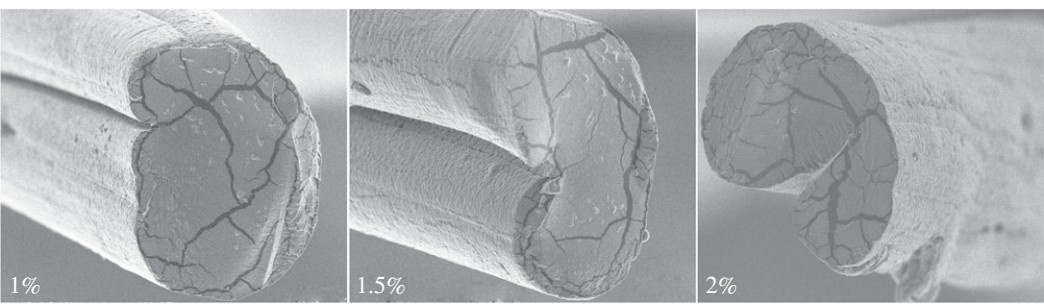

**Figure 14.** SEM images of grooved structure on the fibres of 1, 1.5 and 2% alginate solutions.

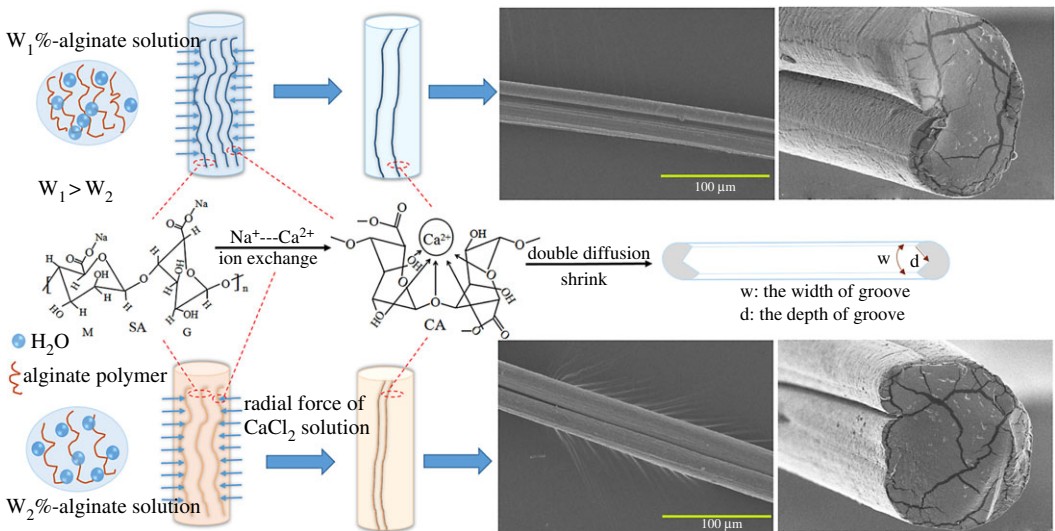

**Figure 15.** Schematic of the formation mechanism of grooves on the fibre.

scarce, resulting in impeding the control of the dimension of grooves. Therefore, the forming process of grooves on the fibres was investigated as follows.

The schematic diagram of the formation mechanism of grooved structure is depicted in figure 15. The linear macromolecular chain of alginate polymers was composed of poly-mannuronate (M) and poly-guluronate (G); an ion-exchange reaction between $Na^+$ of polymer chain and $Ca^{2+}$ of calcium chloride solution was triggered, leading to the formation of alginate fibre with 'egg-box' stereo structure. Various concentrations of alginate solution would cause the difference in the aggregation of polymer, which impacted on the rate of shrink. Moreover, the rate of ion exchange was proportional to the concentration of alginate solution, which also affected the rate of double diffusion. Both the roles of shrink and double diffusion led to the formation of grooves. When alginate solution contacted with $CaCl_2$ solution, $Ca^{2+}$ ion and water molecular of $CaCl_2$ solution diffused into the alginate solution, whereas the same diffusion for $Na^+$ ion of alginate solution. This double diffusion was triggered; meanwhile, the ion exchange between $Na^+$ and $Ca^{2+}$ was also incurred. But, the rates of double diffusion were diverse, which caused the shrinkage of sheath of fibre. Hence, the surface of fibre generated wrinkle and then V-shaped or concave groove was obtained after solvent evaporated. As exhibited in figure 15, the $W_1\%$ and $W_2\%$ were corresponding to 1.5% and 1% alginate solutions, respectively. The higher concentration of alginate solution improved the rate of ion exchange, promoting the double diffusion. That was more beneficial to form the groove on fibres, as shown the bigger grooves of 1.5% alginate solution than that of 1% in the figures.

## 4. Conclusion

Alginate fibres with grooved structure were fabricated via the microfluidic spinning device, consisting of a coaxial flow of two solutions. The influence of flow rate and spinning solution concentration on fibre diameter were investigated in this work, illustrating the minimum spinning concentration being 1%. In addition, the crystal structure and groove depth of fibre prepared by 1, 1.5 and 2% alginate solutions were compared. Ultimately, it could be concluded that the fibre diameter increased with an increase in the core flow rate, whereas on the contrary for sheath flow rate. Fibre diameter and groove depth also demonstrated a positive correlation with the concentration of spinning solution. Both roles of shrink and double diffusion resulted in the generation of wrinkle on the cortex of fibre, and then V-shaped or concave grooves were achieved after the solvent evaporated. Alginate fibres, with topological structure as the substitute, have the capability of mimicking extracellular matrix for promoting the cell and tissue growth, which would be the candidate for the wound dressings or tissue scaffolds.

Data accessibility. Raw data used to create figures 2, 3, 6 and 10 have been uploaded as the electronic supplementary material.

**Authors' contributions.** X.Z. and B.D. contributed to the project ideas and design. X.Z. conducted all the experiments and wrote the manuscript draft. L.W. revised and checked the manuscript. Q.L. and D.L. helped to analysis and discuss the results. All authors gave final approval for publication.

**Competing interests.** We declare we have no competing interests.

**Funding.** This work was supported by 'Postdoctoral Research Funding Scheme' Project from Jiangsu Province and Postdoctoral Research Funding (grant no. 1065210232188490).

**Acknowledgements.** The authors thank all the teachers of Key Laboratory of Eco-Textiles for providing the test instruments.

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
