## [Reviewer comments · Royal Society Open Science]

Review History

RSOS-181928.R0 (Original submission)

Review form: Reviewer 1

Is the manuscript scientifically sound in its present form?

Yes

Are the interpretations and conclusions justified by the results?

Yes

Is the language acceptable?

Yes

Is it clear how to access all supporting data?

Yes

Do you have any ethical concerns with this paper?

No

Have you any concerns about statistical analyses in this paper?

No

Recommendation?

Accept with minor revision (please list in comments)

Comments to the Author(s)

Comments:

This work reported a new fabricated process of alginate microfibers with grooves via microfluidic spinning technique, and further discussed the formation mechanism of this grooved-structure. This work is interesting and have the chances to be published in Royal Society Open Science after minor revision considering the followings.

1. Why the author utilized the microfluidic spinning technology? What are the advantages of microfluidic spinning technology compared with conventional spinning technology?
2. Some important references, which is significant in microfluidic synthesis of fibers should be cited. (Nature Communications, DOI: 10.1038/s41467-018-06914-7. J. Mater. Chem. A, 2018, 6, 8940-8946. Adv. Funct. Mater. 2017, 1702493. Advanced materials 30, 1705082. Advanced Functional materials 25, 2250-2259)

Review form: Reviewer 2

Is the manuscript scientifically sound in its present form?

No

Are the interpretations and conclusions justified by the results?

No

Is the language acceptable?

Yes

Is it clear how to access all supporting data?

Yes

Do you have any ethical concerns with this paper?

No

Have you any concerns about statistical analyses in this paper?

No

Recommendation?

Reject

Comments to the Author(s)

This manuscript was transferred from RSC Advances with previous review comments. While the authors have addressed some concerns in the previous referees' reports, as a new referee I agree with much of the previous referees' comments and feel that the revised manuscript did not address the major concerns.

My major concern is that the main purpose of this study is not clear, therefore dramatically dilutes the novelty of the report in its current format. The authors acknowledge in the response letter that MST is a well characterized and widely reported technique, and the novelty in the

current study is to include a collector, with the control of rotation rate etc, to form fibers and potential dressing structure with further cross-linking. Including a collector for MST is not impressive, but if that makes big difference, the main content of the report should be how the this new addition has made the fiber differently (better efficiency? new feature?). On the other hand, if the report wants to focus on the grooved feature on individual fibers (as the title hints), the authors should report extensively with experimental data and series discussion on the mechanism of the groove formation (the current discussion is very brief and not convincing without data to support the vague claim of double diffusion, e.g., how the double diffusion create the groove surface?). On the other hand, extensive characterization of the individual fibers and polymer solutions is nothing new and unnecessary. Rather it gives the impression of a lousy study with clumsy data but without careful plan.

Now talking about the potential dressing structure, how will these be different from a previous paper that has fabricated the alginate microfiber with grooves for scaffolds by MST in one step (Advanced Materials 24.31 (2012): 4271-4277)? Obviously, the word groove in that paper and this manuscript means differently (the dressing sheet vs. shrinkage of individual fiber surface). I don't see how the fiber in this report is characterized as 3D as claimed by the authors while the groove sheet in that paper is 2D.

Meanwhile, some more revisions are required as below for possible re-submission:

Q1. The dimension of the microfluidic channels should be measured, which influences the size of alginate fibers and its grooves. Also, the design of spinneret as shown in Figure 1 (c), especially how CaCl₂ and alginate meet, should be stated clearly. This would be one of the possible reasons for grooves formation.

Q2. The authors thought reasons for grooves are the shrinkage of alginate fiber after solvent evaporation and the double diffusion between CaCl₂ and alginate solution. But there is no evidence or experiments to prove it, which is strongly needed.

Q3. In discussion 3.3, the stated working conditions were fuzzy and confusing in each case, which makes conclusion unconvincing. In Figure 4, what were the core and sheath flow rates when the concentration of alginate varied? In Figure 5, what was the concentration of alginate and CaCl₂, and which flow rate was varied as the value shown on images?

Q4. There are problems in figures. There is no discussion about Figure 10 and what does the y axis mean in Figure 10? In Figure 11, all three graphs should have the same axis range that can clearly show the difference. Distinct figure explanation should be applied in Figure 12.

Q5. In authors' design, ethanol was used as solvent for the sheath flow, 3% (w/v) CaCl₂, to induce the gelation of alginate. Therefore there is ethanol residua in the generated alginate fiber. But ethanol should be avoided in applications for cell culture like wound dressing or tissue scaffold because of cytotoxicity. There may be problem in its possible biomedical applications.

Review form: Reviewer 3

Is the manuscript scientifically sound in its present form?

Yes

Are the interpretations and conclusions justified by the results?

Yes

Is the language acceptable?

Yes

Is it clear how to access all supporting data?

Yes

Do you have any ethical concerns with this paper?

No

Have you any concerns about statistical analyses in this paper?

No

Recommendation?

Major revision is needed (please make suggestions in comments)

Comments to the Author(s)

The authors represent a way to fabricate alginate hydrogel fiber based on concentration and flow rate. However, the data was not represented and poorly organized. The author should largely improve the format.

1. The author mentioned the viscosity dependent fabrication of microfiber. From Figure 5a, it looks like these parameters affect the cross-section morphology of fibers. Can the author show a cross-section projection image, how it was being affected by concentrations and flow rates.
2. In figure 6 caption, the author mentioned the "curvature of fiber diameter", but in the figure, it is only the diameter. I just wonder what will be the curvature. And does the curvature on the microfiber is uniform?
3. What does Figure 7 provide? it looks like it just saying "the alginate fiber is alginate"
4. The same story with XRD. What does it provide?
5. The authors claimed that they have a groove structure on fiber from Figure 9. But their scale is even in tens of micron. We can not see any micron or submicron structure. SEM can easily go to sub micron and nanometers, please do a better image of fibers.
6. Figure 10 has a similar information to Figure 6.
7. Does AFM image on Figure 11 only show one groove on the fiber? But that scale is too large for a single groove. If it is the fiber? It is too small for a single fiber shown in Figure 10.

Decision letter (RSOS-181928.R0)

01-Mar-2019

Dear Dr Zhang:

Title: Facile fabrication and characterization on alginate microfibers with grooved structure via the microfluidic spinning

Manuscript ID: RSOS-181928

The editor assigned to your manuscript has now received comments from reviewers. We would like you to revise your paper in accordance with the referee and Subject Editor suggestions which can be found below (not including confidential reports to the Editor). Please note this decision does not guarantee eventual acceptance.

Please submit your revised paper before 24-Mar-2019. Please note that the revision deadline will expire at 00.00am on this date. If we do not hear from you within this time then it will be assumed that the paper has been withdrawn. In exceptional circumstances, extensions may be possible if agreed with the Editorial Office in advance. We do not allow multiple rounds of revision so we urge you to make every effort to fully address all of the comments at this stage. If deemed necessary by the Editors, your manuscript will be sent back to one or more of the original reviewers for assessment. If the original reviewers are not available we may invite new reviewers.

Please also include the following statements alongside the other end statements. As we cannot publish your manuscript without these end statements included, if you feel that a given heading is not relevant to your paper, please nevertheless include the heading and explicitly state that it is not relevant to your work.

- Ethics statement

Please clarify whether you received ethical approval from a local ethics committee to carry out your study. If so please include details of this, including the name of the committee that gave consent in a Research Ethics section after your main text. Please also clarify whether you received informed consent for the participants to participate in the study and state this in your Research Ethics section.

OR

Please clarify whether you obtained the necessary licences and approvals from your institutional animal ethics committee before conducting your research. Please provide details of these licences and approvals in an Animal Ethics section after your main text.

OR

Please clarify whether you obtained the appropriate permissions and licences to conduct the fieldwork detailed in your study. Please provide details of these in your methods section.

Yours sincerely,
Dr Laura Smith

Publishing Editor, Journals

RSC Associate Editor:
Comments to the Author:
(There are no comments.)

RSC Subject Editor:
Comments to the Author:
(There are no comments.)

Reviewers' Comments to Author:
Reviewer: 1

Comments to the Author(s)
Comments:

This work reported a new fabricated process of alginate microfibers with grooves via microfluidic spinning technique, and further discussed the formation mechanism of this grooved-structure. This work is interesting and have the chances to be published in Royal Society Open Science after minor revision considering the followings.

1. Why the author utilized the microfluidic spinning technology? What are the advantages of microfluidic spinning technology compared with conventional spinning technology?
2. Some important references, which is significant in microfluidic synthesis of fibers should be cited. (Nature Communications, DOI: 10.1038/s41467-018-06914-7. J. Mater. Chem. A, 2018, 6, 8940-8946. Adv. Funct. Mater. 2017, 1702493. Advanced materials 30, 1705082. Advanced Functional materials 25, 2250-2259)

Reviewer: 2

Comments to the Author(s)

This manuscript was transferred from RSC Advances with previous review comments. While the authors have addressed some concerns in the previous referees' reports, as a new referee I agree with much of the previous referees' comments and feel that the revised manuscript did not address the major concerns.

My major concern is that the main purpose of this study is not clear, therefore dramatically dilutes the novelty of the report in its current format. The authors acknowledge in the response letter that MST is a well characterized and widely reported technique, and the novelty in the current study is to include a collector, with the control of rotation rate etc, to form fibers and potential dressing structure with further cross-linking. Including a collector for MST is not

impressive, but if that makes big difference, the main content of the report should be how the this new addition has made the fiber differently (better efficiency? new feature?). On the other hand, if the report wants to focus on the grooved feature on individual fibers (as the title hints), the authors should report extensively with experimental data and series discussion on the mechanism of the groove formation (the current discussion is very brief and not convincing without data to support the vague claim of double diffusion, e.g., how the double diffusion create the groove surface?). On the other hand, extensive characterization of the individual fibers and polymer solutions is nothing new and unnecessary. Rather it gives the impression of a lousy study with clumsy data but without careful plan.

Now talking about the potential dressing structure, how will these be different from a previous paper that has fabricated the alginate microfiber with grooves for scaffolds by MST in one step (Advanced Materials 24.31 (2012): 4271-4277)? Obviously, the word groove in that paper and this manuscript means differently (the dressing sheet vs. shrinkage of individual fiber surface). I don't see how the fiber in this report is characterized as 3D as claimed by the authors while the groove sheet in that paper is 2D.

Meanwhile, some more revisions are required as below for possible re-submission:

Q1. The dimension of the microfluidic channels should be measured, which influences the size of alginate fibers and its grooves. Also, the design of spinneret as shown in Figure 1 (c), especially how CaCl₂ and alginate meet, should be stated clearly. This would be one of the possible reasons for grooves formation.

Q2. The authors thought reasons for grooves are the shrinkage of alginate fiber after solvent evaporation and the double diffusion between CaCl₂ and alginate solution. But there is no evidence or experiments to prove it, which is strongly needed.

Q3. In discussion 3.3, the stated working conditions were fuzzy and confusing in each case, which makes conclusion unconvincing. In Figure 4, what were the core and sheath flow rates when the concentration of alginate varied? In Figure 5, what was the concentration of alginate and CaCl₂, and which flow rate was varied as the value shown on images?

Q4. There are problems in figures. There is no discussion about Figure 10 and what does the y axis mean in Figure 10? In Figure 11, all three graphs should have the same axis range that can clearly show the difference. Distinct figure explanation should be applied in Figure 12.

Q5. In authors' design, ethanol was used as solvent for the sheath flow, 3% (w/v) CaCl₂, to induce the gelation of alginate. Therefore there is ethanol residua in the generated alginate fiber. But ethanol should be avoided in applications for cell culture like wound dressing or tissue scaffold because of cytotoxicity. There may be problem in its possible biomedical applications.

Reviewer: 3

Comments to the Author(s)

The authors represent a way to fabricate alginate hydrogel fiber based on concentration and flow rate. However, the data was not represented and poorly organized. The author should largely improve the format.

1. The author mentioned the viscosity dependent fabrication of microfiber. From Figure 5a, it looks like these parameters affect the cross-section morphology of fibers. Can the author show a cross-section projection image, how it was being affected by concentrations and flow rates.

2. In figure 6 caption, the author mentioned the “curvature of fiber diameter”, but in the figure, it is only the diameter. I just wonder what will be the curvature. And does the curvature on the microfiber is uniform?
3. What does Figure 7 provide? it looks like it just saying “the alginate fiber is alginate”
4. The same story with XRD. What does it provide?
5. The authors claimed that they have a groove structure on fiber from Figure 9. But their scale is even in tens of micron. We can not see any micron or submicron structure. SEM can easily go to sub micron and nanometers, please do a better image of fibers.
6. Figure 10 has a similar information to Figure 6.
7. Does AFM image on Figure 11 only show one groove on the fiber? But that scale is too large for a single groove. If it is the fiber? It is too small for a single fiber shown in Figure 10.

Author's Response to Decision Letter for (RSOS-181928.R0)

See Appendix A.

RSOS-181928.R1 (Revision)

Review form: Reviewer 1

Is the manuscript scientifically sound in its present form?

Yes

Are the interpretations and conclusions justified by the results?

Yes

Is the language acceptable?

Yes

Is it clear how to access all supporting data?

Yes

Do you have any ethical concerns with this paper?

No

Have you any concerns about statistical analyses in this paper?

No

Recommendation?

Accept as is

Comments to the Author(s)

The authors have answered the raised questions and the present revision is suggested to be accepted.

Review form: Reviewer 2

Is the manuscript scientifically sound in its present form?

Yes

Are the interpretations and conclusions justified by the results?

Yes

Is the language acceptable?

Yes

Is it clear how to access all supporting data?

Yes

Do you have any ethical concerns with this paper?

No

Have you any concerns about statistical analyses in this paper?

No

Recommendation?

Accept as is

Comments to the Author(s)

The authors have spent great effort to improve the manuscript, and it is in a good shape to be published at its current format.

Decision letter (RSOS-181928.R1)

15-Apr-2019

Dear Dr zhang:

Title: Facile fabrication and characterization on alginate microfibers with grooved structure via the microfluidic spinning

Manuscript ID: RSOS-181928.R1

It is a pleasure to accept your manuscript in its current form for publication in Royal Society Open Science. The chemistry content of Royal Society Open Science is published in collaboration with the Royal Society of Chemistry.

Reviewer(s)' Comments to Author:
Reviewer: 1

Comments to the Author(s)
The authors have answered the raised questions and the present revision is suggested to be accepted.

Reviewer: 2

Comments to the Author(s)
The authors have spent great effort to improve the manuscript, and it is in a good shape to be published at its current format.

Appendix A

Dear editor-Professor Laura Smith :

Thanks very much for your letter and the comments from all the reviewers about the manuscript # RSOS-181928 entitled “Facile fabrication and characterization on alginate microfibers with grooved structure via the microfluidic spinning”. The manuscript has been revised according to reviewers’ comments, carefully and accordingly.

I appreciate the reviewers’ careful review, professional comments and precious suggestion. I have tried my best to read the comments carefully again and again, then did some corresponding amendments in the manuscript and provided some supplementary file containing some added experiments. We have learned much from the reviewers’ comments, which are professional, encouraging and constructive. A point to point response to the comments from each reviewer have been edited listed as following.

Thank you for your suggestion again, meaning the recognition and encouragement to me. The manuscript has been much improved after revising according to the comments and suggestions from all the reviewers.

Thanks again, and I hope that I can learn much more knowledge from you.

If you have any question about this paper, please don’t hesitate to contact me.

Best wishes,

List of Actions

LOA: Introduction

LOA 1: “Some important references, which is significant in microfluidic synthesis of fibers should be cited. (Nature Communications, DOI: 10.1038/s41467-018-06914-7. J. Mater. Chem. A, 2018, 6, 8940–8946. Adv. Funct. Mater. 2017, 1702493. Advanced materials 30, 1705082. Advanced Functional materials 25, 2250-2259)”

has been cited in the introduction and highlighted in the manuscript.

[25]X. Wu, Y. Xu, Y. Hu, G. Wu, H. Cheng, Q. Yu, K. Zhang, W. Chen and S. Chen, Nat Commun, 2018, 9, 4573.

[26]X. Wu, G. Wu, P. Tan, H. Cheng, R. Hong, F. Wang and S. Chen, Journal of Materials Chemistry A, 2018, 6, 8940-8946.

[27]G. Wu, P. Tan, X. Wu, L. Peng, H. Cheng, C.-F. Wang, W. Chen, Z. Yu and S. Chen, Advanced Functional Materials, 2017, 27, 1702493.

[28]R. Xie, P. Xu, Y. Liu, L. Li, G. Luo, M. Ding and Q. Liang, Adv Mater, 2018, 30, e1705082.

[29]X. Shi, S. Ostrovidov, Y. Zhao, X. Liang, M. Kasuya, K. Kurihara, K. Nakajima, H. Bae, H. Wu and A. Khademhosseini, Advanced Functional Materials, 2015, 25, 2250-2259.

LOA: Results and discussions

LOA 2: Discussion 3.1: “How CaCl₂ and alginate meet as shown in Figure 1 (c)?” has been added and highlighted in the manuscript.

As shown in Figure 1(c), alginate solution and CaCl₂ solution was orderly pumped into syringe, PTFE tubes and microfluidic channels, then both solutions in the end of micro-channels would meet to form the fibers. That was the result of chelation between the carboxyl group of alginate molecular chain and the calcium ion of CaCl₂ solution to make alginate form the “egg-box” structure. The buckled chain of guluronic acid units acting as a two-dimensional analogue of corrugated egg-box with interstices in the calcium ions packed and coordinated to form fiber.

LOA 3: Discussion 3.1: “The dimension of the microfluidic channels?” has been added and highlight in the manuscript.

Spinning solution consisted of alginate solution as the core flow and CaCl₂ solution acted as the sheath flow, were respectively loaded into two syringes (10 ml) mounted a 23 G and 17G stainless steel needle. The diameter of core and sheath channel was 340 μm and 490 μm, respectively.

LOA 4: Discussion 3.3: “In Figure 4, what were the core and sheath flow rates when the concentration of alginate varied?” has been added and highlighted in the

manuscript.

Morphological structure of fibers fabricated by various alginate concentration were investigated as displayed in Figure 4, with core and sheath flow rates being 1.5 mL/min and 3 mL/min.

LOA 5: Discussion 3.3: “In Figure 5, what was the concentration of alginate and CaCl₂? ” has been added and highlighted in the manuscript.

Morphological structure of fibers fabricated by diverse flow rates were investigated as displayed in Figure 5, with the concentration of alginate and CaCl₂ solution being 2% and 3%.

LOA 6: Discussion 3.3: “In Figure 5: which flow rate was varied as the value shown on images? ” has been added and highlighted in the manuscript.

The diameter of fiber increased from $31.98 \pm 0.83 \mu\text{m}$ to $47.72 \pm 1.15 \mu\text{m}$ as the core and sheath flow rates varied from 1.5 mL/min to 6.5 mL/min, as shown in Figure 5.

LOA 7: Discussion 3.3: “Can the author show a cross-section projection image, how it was being affected by concentrations and flow rates? ” has been added and highlighted in the manuscript.

The morphology structure of alginate fiber clearly demonstrated that polymer concentration and flow rates not only affected the fibrous diameter, but also acted as the key parameter to the processing. Hence, the cross-section morphology of fibers fabricated by various concentration of alginate solution and flow rates were investigated. As shown in Figure 7 and 8, the cross-section of fibers was circular with a sunken segment and the grooves could be observed, which was affected by the alginate concentration and flow rates. The dimension of grooves increased with an increasing in the alginate solution and flow rates under a certain condition, and then exhibited a decreased tendency. The bigger alginate concentration and flow rates improved the rate of double diffusion, resulting in the clear grooves. However, fibers formed quickly with the concentration and flow rates being high, impeding the double diffusion. That was not beneficial to the formation of grooves.

Figure 7. Cross-section images of fibers fabricated by various alginate solution

Figure 8. Cross-section images of fibers fabricated at diverse flow rates

LOA 8: Discussion 3.6: “Characterization of the fibers diameter and surface structure” has been amended and highlighted in the manuscript.

The surface structure and diameter distribution of fibers fabricated by 1%, 1.5% and 2% alginate solution were analyzed by SEM and Image Pro Plus software, as described in Figure 11 and Figure 12. It could be clearly observed that the surface of all the fibers exhibited grooved structure and the fiber fabricated with higher alginate

concentration demonstrated bigger depth and width of grooves, as depicted in Figure 11. The rate of ion-exchange between Ca^{2+} and Na^+ was proportional to concentration of polymer, impacting on the formation of fibers and grooves. The higher concentration resulted in the larger ion-exchange rate, inducing the faster forming process of grooves and then generating the bigger dimension of grooves.

LOA 9: Discussion 3.6: “There is no discussion about Figure 10?” has been corrected and highlighted in the manuscript.

The diameter distribution of fibers fabricated by the concentration of 1%, 1.5% and 2% alginate were exhibited in Figure 12. It could be found that the diameter of 1% alginate fiber was chiefly between 23 μm and 26 μm , whereas that of 1.5% alginate fiber being mainly around 27 μm . For 2% alginate fiber, the diameter was widespread, ranging from 29 μm to 32 μm . It could be clearly found that the diameter distribution of fibers was proportional to the alginate concentration. Overall, the fibrous diameter exhibited an increasing tendency with increasing the alginate polymer solution.

Figure 12. Diameter distribution of fibers spun at (a) 1%, (b) 1.5% and (c) 2% alginate solutions

LOA 10: Discussion 3.6: “The authors claimed that they have a groove structure on fiber from Figure 9. But their scale is even in tens of micron. We can not see any micron or submicron structure. SEM can easily go to sub micron and nanometers, please do a better image of fibers?” has been corrected and highlighted in the manuscript.

Figure 11. SEM image of groove with diverse width and depth on the fiber spun at (a) 1%, (b) 1.5% and (c) 2% alginate solutions

LOA 11: Discussion 3.6: “There is no discussion about Figure 10?” has been added and highlighted in the manuscript.

The diameter distribution of fibers fabricated by the concentration of 1%, 1.5% and 2% alginate were exhibited in Figure 10. It could be found that the diameter of 1% alginate fiber was chiefly between 23 μm and 26 μm , whereas that of 1.5% alginate fiber being mainly around 27 μm . For 2% alginate fiber, the diameter was widespread, ranging from 29 μm to 32 μm . It could be clearly found that the diameter distribution of fibers was proportional to the alginate concentration. Overall, the fibrous diameter exhibited an increasing tendency with increasing the alginate polymer solution.

LOA 12: Discussion 3.7: “Does AFM image on Figure 11 only show one groove on the fiber? But that scale is too large for a single groove. If it is the fiber? It is too small for a single fiber shown in Figure 10?” has been corrected and highlighted in the manuscript.

The microgroove depths of 1% and 1.5% alginate fiber was 278.37 ± 2.23 nm and 683.69 ± 3.18 nm respectively, whilst that of 2% alginate fiber being 727.52 ± 3.52 nm. Furthermore, the widths of groove for 1%, 1.5% and 2% samples was 251.33 ± 1.67 nm, 419.67 ± 2.82 nm and 730.67 ± 3.12 nm, respectively. AFM image on Figure 11 exhibited one groove on the fibers, and the scan scale was between 4 μm and 10 μm , which was determined by the width and depth of grooves. And the diameter distribution of fibers was majorly between 20 μm and 32 μm .

LOA 13: Discussion 3.7: “the cross-section images of fibers with grooved structure” has been added and highlighted in the manuscript.

Figure 13. SEM images of grooved structure on the fibers of 1%, 1.5% and 2% alginate solution

LOA 14: Discussion 3.8: “Distinct figure explanation should be applied in Figure 12” has been added and highlighted in the manuscript.

As exhibited in Figure 15, the $W_1\%$ and $W_2\%$ was corresponding to 1.5% and 1% alginate solution. The higher concentration of alginate solution improved the rate of ion-exchange, promoting the double diffusion. That was more beneficial to form the groove on fibers, as shown the bigger grooves of 1.5% alginate solution than that of 1% in the following figures.

Reviewers' Comments to Author:

Reviewer: 1

Dear reviewer 1:

Many thanks for your careful review and professional comments. I have learnt much from that, which do much favor for my study. I have tried my best to revise the manuscript and provided the supplementary file including some added experiments.

I appreciate your affirmation and encouragement to my work. I have learned much from the precious suggestion you offered, which are professional, precise and constructive. It is very significant to improve my research level and find many defects in my study. And I promise I will try my best to work harder. In addition, I have made some modification listed as following and revised in the manuscript.

Thanks again, and I hope that I can learn much more knowledge from you.

Best regards,

Comments to the Author(s)

Comments:

This work reported a new fabricated process of alginate microfibers with grooves via microfluidic spinning technique, and further discussed the formation mechanism of this grooved-structure. This work is interesting and have the chances to be published in Royal Society Open Science after minor revision considering the followings.

1. Why the author utilized the microfluidic spinning technology? What are the advantages of microfluidic spinning technology compared with conventional spinning technology?

As we know, melt spinning, wet spinning, and electrospinning are common methods to produce microfibers. However, melt spinning needs bulky and heavy equipment for its high temperature process, while the electrospinning involves volatile organic solvents, rendering them unacceptable for the fabrication of wound dressing. Especially, the alginate couldn't be fabricated by the water as solvent via the electrospinning technology. The process of wet spinning was very complicate and old, which was not facilitate to fabricate the fibers with special structure as the core-shell, hollow, double-layered and grooved structure, et al. However, the microfluidic spinning technology is simple, cost-effective and compatible with biological materials and thus an alternative method for producing the fibers with many different structure. In addition, the microfluidic spinning devices design is simple, flexibility and efficient to fabricate grooved microfibers using a one-step continuous process. The diameters of microfibers and grooves could be tuned by adjusting the spinning parameters. Thus, it was the best choice that the microfluidic spinning technology was used to fabricate the alginate fibers with grooved structure.

2. Some important references, which is significant in microfluidic synthesis of fibers should be cited. (Nature Communications, DOI: 10.1038/s41467-018-06914-7. J. Mater. Chem. A, 2018, 6, 8940–8946. Adv. Funct. Mater. 2017, 1702493. Advanced materials 30, 1705082. Advanced Functional materials 25, 2250-2259)

Many thanks for the suggested references and I have cited in the manuscript.

[25]X. Wu, Y. Xu, Y. Hu, G. Wu, H. Cheng, Q. Yu, K. Zhang, W. Chen and S. Chen, Nat Commun, 2018, 9, 4573.

[26]X. Wu, G. Wu, P. Tan, H. Cheng, R. Hong, F. Wang and S. Chen, Journal of Materials Chemistry A, 2018, 6, 8940-8946.

[27]G. Wu, P. Tan, X. Wu, L. Peng, H. Cheng, C.-F. Wang, W. Chen, Z. Yu and S. Chen, Advanced Functional Materials, 2017, 27, 1702493.

[28]R. Xie, P. Xu, Y. Liu, L. Li, G. Luo, M. Ding and Q. Liang, Adv Mater, 2018, 30, e1705082.

[29]X. Shi, S. Ostrovidov, Y. Zhao, X. Liang, M. Kasuya, K. Kurihara, K. Nakajima, H. Bae, H. Wu and A. Khademhosseini, Advanced Functional Materials, 2015, 25, 2250-2259.

Reviewer: 2

Dear the Reviewer 2:

I appreciate your careful review, professional comments and precious suggestion, which is beneficial to my research. I have tried my best to read the comments carefully again and again, then did some corresponding amendments in the manuscript.

Thank you for your valuable advice again. It is very significant to improve my research level and help me to find many shortcomings in my study. Please let me know if there are some shortages need to be improved. I promise that I will try to work hard and improve the research level.

Thanks again, and I hope that I can learn much more knowledge from you.

If you have any question about this paper, please don't hesitate to contact me.

Best wishes,

Comments to the Author(s)

My major concern is that the main purpose of this study is not clear, therefore dramatically dilutes the novelty of the report in its current format. The authors acknowledge in the response letter that MST is a well characterized and widely reported technique, and the novelty in the current study is to include a collector, with the control

of rotation rate etc, to form fibers and potential dressing structure with further cross-linking. Including a collector for MST is not impressive, but if that makes big difference, the main content of the report should be how this new addition has made the fiber differently (better efficiency? new feature?). On the other hand, if the report wants to focus on the grooved feature on individual fibers (as the title hints), the authors should report extensively with experimental data and series discussion on the mechanism of the groove formation (the current discussion is very brief and not convincing without data to support the vague claim of double diffusion, e.g., how the double diffusion create the groove surface?). On the other hand, extensive characterization of the individual fibers and polymer solutions is nothing new and unnecessary. Rather it gives the impression of a lousy study with clumsy data but without careful plan.

This manuscript was to fabricate the alginate fibers with groove by the microfluidic spinning technology (MST), characterize the structure and disclose the forming mechanism of grooves. The MST was better efficiency to fabricate fiber dressing or fabric dressing in one step by various collector with adjustable rotation rate and movement velocity. Furthermore, MST facilitate to fabricate fibers with new feature including hollow, double-layered, core-shell and grooved structure et al. To investigate the grooves on the fiber surface better, some supplementary experiment about the grooves and some study on the fabrication of fibers and grooves have been added and highlighted in the manuscript.

Now talking about the potential dressing structure, how will these be different from a previous paper that has fabricated the alginate microfiber with grooves for scaffolds by MST in one step (*Advanced Materials* 24.31 (2012): 4271-4277)? Obviously, the word groove in that paper and this manuscript means differently (the dressing sheet vs. shrinkage of individual fiber surface). I don't see how the fiber in this report is characterized as 3D as claimed by the authors while the groove sheet in that paper is 2D.

That paper (*Advanced Materials* 24.31 (2012): 4271-4277) fabricated the fibers with grooves via the design of microchannel. A PDMS slit channel with microgrooves was fabricated using standard soft lithography methods. The co-axial grooved slit

channel was constructed by aligning and bonding two grooved PDMS channels by oxygen plasma treatment. However, this manuscript fabricated the fibers with grooves by adjust the solution concentration and flow rates. Grooved structure was analyzed at three dimensional point including the groove width and depth, and grooves on the fiber surface to form three dimensional structure. But, after I read the comments carefully and considered the question put forward by the reviewer again and again, I agreed with the reviewer and thought the fibers being not characterized as 3D materials. Thus, I have corrected it in the manuscript.

Q1. The dimension of the microfluidic channels should be measured, which influences the size of alginate fibers and its grooves. Also, the design of spinneret as shown in Figure 1 (c), especially how CaCl₂ and alginate meet, should be stated clearly. This would be one of the possible reasons for grooves formation.

The diameters of core and sheath channels was 340 μm and 490 μm, respectively. As shown in Figure 1(c), alginate solution and CaCl₂ solution was orderly pumped into syringe, PTFE tubes and microfluidic channels, then both solutions in the end of micro-channels would meet to form the fibers. That was the result of chelation between the carboxyl group of alginate molecular chain and the calcium ion of CaCl₂ solution to make alginate form the “egg-box” structure. The buckled chain of guluronic acid units acting as a two-dimensional analogue of corrugated egg-box with interstices in the calcium ions packed and coordinated to form fiber.

Q2. The authors thought reasons for grooves are the shrinkage of alginate fiber after solvent evaporation and the double diffusion between CaCl₂ and alginate solution. But there is no evidence or experiments to prove it, which is strongly needed.

Both the roles of shrink and double diffusion led to the formation of grooves. When alginate solution contacted with CaCl₂ solution, Ca²⁺ ion and water molecular of CaCl₂ solution diffused into the alginate solution whereas the Na⁺ ion of alginate solution diffused into CaCl₂ solution. This double diffusion resulted in the ion-exchange between Na⁺ of alginate and Ca²⁺ of CaCl₂ solution to form alginate fiber. However, the diffusion rates of ions were different and the solvent evaporated, which caused the shrink of fibrous cortex. Hence, the surface of fiber generated wrinkle and then V-

shaped or concaved groove was obtained, as exhibited in the image of SEM. That could be proved by the analysis of Ca and Na element content of grooves with EDS, as displayed in Figure S1. It could be calculated that the ratio of Ca and Na element content of alginate fiber was 76.38% and 16.13% respectively, indicating the double diffusion between CaCl_2 and alginate, and the bigger diffusion rate of Ca than that of Na.

Figure S1 EDS image of alginate fiber with grooves

Q3. In discussion 3.3, the stated working conditions were fuzzy and confusing in each case, which makes conclusion unconvincing. In Figure 4, what were the core and sheath flow rates when the concentration of alginate varied? In Figure 5, what was the concentration of alginate and CaCl_2 , and which flow rate was varied as the value shown on images?

In Figure 4, the core and sheath flow rates was 1.5 mL/min and 3 mL/min respectively, when the concentration of alginate varied. In Figure 5, the concentration of alginate and CaCl_2 was 2% and 3% respectively, both core and sheath flow rates were varied as the value shown on images.

Q4. There are problems in figures. There is no discussion about Figure 10 and what does the y axis mean in Figure 10? In Figure 11, all three graphs should have the same axis range that can clearly show the difference. Distinct figure explanation should be applied in Figure 12.

The y axis meant the distribution of fibers diameters. The diameter distribution of fibers fabricated by the concentration of 1%, 1.5% and 2% alginate were exhibited in Figure 10 by the Image Pro Plus software. It could be found that the diameter of 1% alginate fiber was chiefly between 23 μm and 26 μm , whereas that of 1.5% alginate

fiber being mainly around 27 μm . For 2% alginate fiber, the diameter was widespread, ranging from 29 μm to 32 μm . It could be concluded that the diameter distribution of fibers was proportional to the alginate concentration. Overall, the fibrous diameter exhibited an increasing tendency with increasing the alginate polymer solution.

Figure 10. Diameter distribution of fibers spun at (a) 1%, (b) 1.5% and (c) 2% alginate solutions

Figure 11 was marked as Figure 14. Three AFM images displayed different axis range in Figure 14, because the probe used to scan the specimen at a set range to ensure the probe work well. Various width and depth of grooves should be with a mixed scanning range, resulting in the different axis range in the AFM graphs. If all the specimen were scanned at the same axis range, the probe would be broke off when it scanned various width and depth of grooves. Though, the axis range were different in Figure 14, AFM was to measure the depth and width of grooves. I promise that I will try my best to keep the same axis range as possible in the future work, which is more beneficial to analyze and compare the datum.

Figure 12 was marked as Figure 15. The comments “Distinct figure explanation should be applied in Figure 12” put forward by the reviewer has been added and highlighted in the manuscript, as following: As exhibited in Figure 15, the $W_1\%$ and $W_2\%$ was corresponding to 1.5% and 1% alginate solution respectively. The higher concentration of alginate solution improved the rate of ion-exchange, promoting the double diffusion. That was more beneficial to form the groove on fibers, as shown the bigger grooves of 1.5% alginate solution than that of 1% in Figure 15.

Q5. In authors' design, ethanol was used as solvent for the sheath flow, 3% (w/v) CaCl_2 , to induce the gelation of alginate. Therefore there is ethanol residua in the generated alginate fiber. But ethanol should be avoided in applications for cell culture like wound

dressing or tissue scaffold because of cytotoxicity. There may be problem in its possible biomedical applications.

Ethanol should be avoided in application for cell culture like wound dressings or tissue scaffold because of cytotoxicity. This question remind me that the nontoxicity is significant for biomaterials and I will try my best to avoid choosing the cytotoxicity solution. Although, ethanol used as solvent for the fabrication of alginate fiber and it was all gone after a few days, which was proved as the research as previously done (RSC Advs, 2017, 7(62): 39349-39358).

Reviewer: 3

Dear Reviewer 3,

Thanks for your valuable advice, which is beneficial to my research. After read the comments you put forward again and again, I have tried my best to add some experiments and did some improvement as follow. The cross-section of fiber varied with flow rates and concentrations have been added and highlighted in the manuscript, and sub micro or nanometers of SEM images of grooves have been carried out again and corrected in the manuscript. I promise that I will try to work hard and improve the research level.

Thanks again, and I hope that I can learn much more knowledge from you.

If you have any question about this paper, please don't hesitate to contact me.

Best regards,

Comments to the Author(s)

The authors represent a way to fabricate alginate hydrogel fiber based on concentration and flow rate. However, the data was not represented and poorly organized. The author should largely improve the format.

1. The author mentioned the viscosity dependent fabrication of microfiber. From Figure

5a, it looks like these parameters affect the cross-section morphology of fibers. Can the author show a cross-section projection image, how it was being affected by concentrations and flow rates.

The morphology structure of alginate fiber clearly demonstrated that polymer concentration and flow rates not only affected the fibrous diameter, but also acted as the key parameter to the processing. Hence, the cross-section morphology of fibers fabricated by various concentration of alginate solution and flow rates were investigated. As shown in Figure 7 and 8, the cross-section of fibers was circular with a sunken segment and the grooves could be observed, which was affected by the alginate concentration and flow rates. The dimension of grooves increased with an increasing in the alginate solution and flow rates under a certain condition, and then exhibited a decreased tendency. The bigger alginate concentration and flow rates improved the rate of double diffusion, resulting in the clear grooves. However, fibers formed quickly with the concentration and flow rates being high, impeding the double diffusion. That was not beneficial to the formation of grooves, corresponding to the grooves change displayed in the cross-section images of Figure 7 and 8.

Figure 7. Cross-section images of fibers fabricated by various alginate solution

Figure 8. Cross-section images of fibers fabricated at diverse flow rates

2. In figure 6 caption, the author mentioned the “curvature of fiber diameter”, but in the figure, it is only the diameter. I just wonder what will be the curvature. And does the curvature on the microfiber is uniform?

I’m so sorry that the discussion in figure 6 caption mislead you. I just study the effect of flow rates on the fibers diameters and the curves of fiber diameter were displayed in figure 6. The curves of fiber diameter was not the curvature of fiber diameter, and the curvature of fiber diameter was not the research point for this work. But I am going to investigate the curvature of various fiber diameter in next research. Anyway, I’m appreciate this question you proposed, which must be helpful for my research in future.

3. What does Figure 7 provide? it looks like it just saying “the alginate fiber is alginate“

4. The same story with XRD. What does it provide?

Figure 7 and 8 was marked as Figure 9 and 10, respectively. FTIR spectra of alginate polymer and fibers fabricated with diverse concentration of alginate solution in Figure 9 (a) and (b). Figure 9 (a) proved that the alginate polymer and fiber exhibited the similar absorption bands, and there were a slight variation of some peaks intensity because of the egg-box structure of alginate fibers. But, there were no additional new peaks as displayed in Figure 9 (a), indicating the unchanged molecular chain structure.

Figure 9 (b) illustrated the alginate solution concentration had little influence on the chemical groups of fibers, resulting in the similar FTIR spectra of all fibers. Thus, few noticeable shifts of absorption bands were observed in Figure 9, implying the alginate solution concentration impacting on the groove structure whereas no generation of new groups or chemical reaction.

The section in X-ray diffraction were investigated the crystal structure of alginate fibers and polymer, and the degree of crystallinity were exhibited in Figure 10 (a) and (b). Figure 10 (a) illustrated the alginate polymer and fibers displayed the same $2\theta=23.60$ in the spectra, indicating the same phase composition agreed with the analysis of FTIR spectra. But the crystallinity of fiber was higher than that of polymer, which was the tanglesome roll structure of polymer turned to the straight chain structure in the forming process of fibers. The crystallinity of fibers fabricated by 1%, 1.5% and 2% alginate concentration demonstrated a similar behavior in Figure 10 (b), verifying the little influence of concentration on the crystal behavior. Hence, the analysis of XRD was further validated that there was no generation of new chemical groups during the formation of alginate fibers, compared with the polymer. Furthermore, it was also demonstrated that the alginate concentration impacted on the grooves dimension.

5. The authors claimed that they have a groove structure on fiber from Figure 9. But their scale is even in tens of micron. We can not see any micron or submicron structure. SEM can easily go to submicron and nanometers, please do a better image of fibers.

After read the comments carefully, the suggestion put forward by reviewer was absolutely correct and I agree with more. Hence, the groove structure on fibers fabricate by 1%, 1.5% and 2% alginate concentration was scanned again by SEM, as shown in Figure 11. To observe the clearer groove structure, the scale was set as displayed in the figure.

Figure 11. SEM image of groove with diverse width and depth on the fiber spun at (a) 1%, (b) 1.5% and (c) 2% alginate solutions

6. Figure 10 has a similar information to Figure 6.

Figure 10 was marked as Figure 12. Both Figure 6 and 12 investigated the fibers diameter. The variation tendency of fiber diameters with flow rates were displayed in Figure 6, and the distribution of fiber diameters were exhibited in Figure 12. The discussion in Figure 6 was aimed to illustrate the influence of core and sheath flow rate on fibers diameters and provided the change tendency of fiber diameter. But, the discussion in Figure 12 was to explore the effect of alginate concentration on fiber diameters and the diameters distribution of fiber fabricated with 1%, 1.5% and 2% alginate solution were exhibited. The fiber diameters were studied in both figures, whereas the different research point and objective in both figures.

7. Does AFM image on Figure 11 only show one groove on the fiber? But that scale is too large for a single groove. If it is the fiber? It is too small for a single fiber shown in Figure 10.

AFM image on Figure 11 was marked as Figure 14. I'm so sorry that the unit of grooves depth and width in the discussion of AFM analysis from Figure 14 made a mistake, and I have corrected as "The microgroove depths of 1% and 1.5% alginate fiber was 278.37 ± 2.23 nm and 683.69 ± 3.18 nm respectively, whilst that of 2% alginate fiber being 727.52 ± 3.52 nm. Furthermore, the widths of groove for 1%, 1.5% and 2% samples was 251.33 ± 1.67 nm, 419.67 ± 2.82 nm and 730.67 ± 3.12 nm, respectively." AFM image on Figure 14 exhibited one groove on the fibers, and the scan scale was between 4 μ m and 10 μ m, which was determined by the width and depth of grooves. And the diameter distribution of fibers was majorly between 20 μ m and 32 μ m.